# Optimal Learning from Verified Training Data

**Nicholas Bishop**
University of Southampton, UK
nb8g13@soton.ac.uk

**Enrico Gerding**
University of Southampton, UK
eg@ecs.soton.ac.uk

**Long Tran-Thanh**
University of Warwick, UK
long.tran-thanh@warwick.ac.uk

## Abstract

Standard machine learning algorithms typically assume that data is sampled independently from the distribution of interest. In attempts to relax this assumption, fields such as adversarial learning typically assume that data is provided by an adversary, whose sole objective is to fool a learning algorithm. However, in reality, it is often the case that data comes from self-interested agents, with less malicious goals and intentions which lie somewhere between the two settings described above. To tackle this problem, we present a Stackelberg competition model for least squares regression, in which data is provided by agents who wish to achieve specific predictions for their data. Although the resulting optimisation problem is nonconvex, we derive an algorithm which converges globally, outperforming current approaches which only guarantee convergence to local optima. We also provide empirical results on two real-world datasets, the medical personal costs dataset and the red wine dataset, showcasing the performance of our algorithm relative to algorithms which are optimal under adversarial assumptions, outperforming the state of the art.

## 1 Introduction

Many machine learning algorithms perform under the assumption that data is sampled independently and identically from the distribution of interest. In reality, this assumption does not hold in many applications. A canonical example of this is spam email classification. Spammers are constantly adapting and changing the formatting of their emails in an attempt to bypass spam filters. In order to apply machine learning in such settings, approaches such as adversarial machine learning and robust optimisation instead typically assume that data is provided by an adversary who is attempting to fool the learning algorithm. This is a worst-case assumption that is well suited to problems such as spam classification. However, in many scenarios this assumption is overly pessimistic, and, thus, prevents the learning algorithm from achieving high performance (as it always assumes worst-case manipulation).

Instead of such extreme assumptions, we should consider the individuals or organisations that provide data samples and how their motives might influence the data distribution. In many cases, the motivation of data providers is not to hinder generalization but to manipulate the learning algorithm into selecting an outcome which is beneficial for them. In other words, data providers should be viewed as strategic agents seeking to achieve a high payoff. By including the motivations of agents directly in our optimisation process, we can consider a smaller set of possible adversarial manipulations and cease our concerns over adversarial manipulations which are irrational.

Less pessimistic models of data provider manipulation were first considered by Brückner and Scheffer [4], who formulated prediction from manipulated data as a Stackelberg competition. In this framework,

the learner acts as the "leader" by first selecting a learning model with the goal of minimising their own loss. Following this, with full information regarding the model chosen by the learner, data providers modify and submit their data with the aim of minimising their own loss function. The goal of the learner is to select a good model under the assumption that data providers will manipulate their data optimally. Formally, the learner is tasked with solving a bilevel optimisation problem, known as a Stackelberg prediction game (SPG).

Whilst SPGs are nonconvex optimisation problems which are generally NP-hard, it is possible to find local optima via sequential quadratic programming when both the loss functions of the learner and the data provider are convex. However, there is no guarantee that these local optima perform well compared to globally optimal solutions, and thus, existing methods still fail to provide good performance in many scenarios.

To overcome this issue, we present a subclass of SPGs, which include a number of important problems, for which we can find global solutions. In particular, this class consists of least squares linear regression problems in which the learner is interested in predicting one set of outputs whilst data providers are interested in manipulating the learner into predicting another set of outputs. Moreover, we make no assumptions with respect to the output labels of either the learner or the data provider, and, as result, this subclass can be used to model a wide variety of data providers with different motivations.

In more detail, our contributions are as follows: For the first time, we show that solving an SPG of this type is equivalent to solving a quadratically constrained quadratic fractional program with a single constraint using elementary linear algebra. Next, by using a novel combination of Dinkelbach's lemmas for fractional programming [13] and the S-lemma for quadratic programs [3, 19, 25], we show that a combination of bisection search and semidefinite programming can be used to converge to global optima. Finally, we provide numerical results, comparing our algorithm with current approaches. We show empirically that our algorithm outperforms existing methods on multiple real world datasets, halving mean squared error (MSE) in some of our empirical experiments.

## 2 Related Work

Stackelberg competition was originally introduced to model market behaviour and has since been applied in many other contexts. Perhaps the most notable of which being Stackelberg security games (SSGs) [1, 21], in which a defender must commit to a randomised allocation of resources to defend a number of targets from a best responding attacker. Finding the optimal defense strategy is equivalent to solving multiple linear programs [10, 16] and it has been shown that a near-optimal defense strategy can be found in a polynomial number of queries to the attacker [2]. SPGs are similar to SSGs in the sense that they formulate learning an optimal strategy as a bilevel optimisation problem. However, SPG's are tailored specifically to address manipulation of machine learning algorithms.

SPGs themselves have been studied widely since their introduction by Brückner and Scheffer [4]. However, most of the literature focuses on modelling data providers who are either only partly adversarial or are adversarial with limited capabilities. For example, Dong et al. [14] examine an online learning problem akin to spam filtering in which data providers are honest or adversarial at each time step. More recently, Chen et al. [8] provided an online learning algorithm with sublinear Stackelberg regret for binary classification in which data providers may neither be entirely adversarial or honest. In contrast, we consider offline SPGs for linear regression, as opposed to online binary classification. To the best of our knowledge, our paper is the first to consider SPGs for linear least squares regression.

For linear regression, there has been much focus on the development of algorithms which are strategyproof i.e. ensuring that data providers can gain no benefit by manipulating their data. In this vein, Dekel et al. [12] studied a learning scenario in which multiple data providers are selfish and wish to secure the best fitting model for the subset of the training set they hold and gave bounds on the accuracy of any algorithm which is strategyproof in this setting. Many algorithms for strategyproof linear regression are inspired by the strategyproofness of the median for single-peaked preferences [7, 17, 18]. We are not concerned with strategyproofness in our scenario, but only about finding the best performing model regardless of the data manipulation which may take place.

Learning in the presence of noise is closely related to our problem setting. Specifically, worst-case optimisation in the presence of bounded noise is equivalent to minimising loss in the presence of an adversary with limited capability. In particular, El Ghaoui and Lebret [15] proved that minimising

least squares error in the presence of noise bounded in $\ell_2$-norm is equivalent to solving a ridge regression problem with a specific choice of hyperparameter. Similar results also hold for noise bounded in $\ell_1$-norm [26]. In fact, our proof techniques share many similarities with those of El Ghaoui and Lebret [15], including the application of an S-lemma. In contrast, we are looking beyond the worst-case, and the subclass of SPGs we consider allow us to model data providers with more nuanced motivations, which we believe are more likely to arise in practice. More specifically, we consider a setting in which the adversarial assumption, made in the worst-case noise setting, is replaced by the assumption that data providers will manipulate their data with the aim of minimising their own, predetermined, objective function.

## 3 Model

We consider a scenario in which a learner is tasked with choosing a linear predictor, $\mathbf{w} \in \mathbb{R}^n$. Data providers sample tuples of the form $(\mathbf{x}, y, z)$ from a distribution $\mathcal{D}$. The first output, $y \in \mathbb{R}$ is the output label of interest to the learner, while the output label $z \in \mathbb{R}$ is the output label of interest to the data provider, and $\mathbf{x} \in \mathbb{R}^n$ is the input example. We assume the learner has access to a sample, $S = \{(\mathbf{x}_i, y_i, z_i)\}_{i=1}^m$, of unmodified data points which it can use as a training set. For example, such a training sample may be obtained from a reliable, but costly verification process. While it may be too expensive to verify every single data point, the learner may be able to verify a small sample for the purpose of training. After training is completed, the learner must commit to a single linear predictor.

At test time, data providers will continue to sample input examples from the distribution $\mathcal{D}$. With knowledge of the linear predictor chosen by the learner at training time, the data provider is allowed to modify the input examples in each test example. However, the data provider pays a cost of $c(\mathbf{x}, \hat{\mathbf{x}})$ for modifying an input example $\mathbf{x}$ into a different input example $\hat{\mathbf{x}}$. In what follows, we assume that $c(\mathbf{x}, \hat{\mathbf{x}}) = \gamma ||\mathbf{x} - \hat{\mathbf{x}}||_2^2$, where $\gamma > 0$ is a parameter which controls the tradeoff the data provider makes between low cost data manipulations and bringing the learner's prediction as close to $z$ as possible.

The goal of the data provider is to manipulate the learner into predicting $z$ whilst ensuring that the cost of manipulation is low. Formally speaking, the data provider is tasked with solving the following optimisation problem for each test example:

$$\mathbf{x}^* = \operatorname*{argmin}_{\hat{\mathbf{x}}} \left\| \mathbf{w}^T \hat{\mathbf{x}} - z \right\|^2 + \gamma \left\| \mathbf{x} - \hat{\mathbf{x}} \right\|^2$$

Meanwhile, the goal of the learner is to choose a linear predictor which accurately predicts the correct label $y$ for the manipulated data point $\hat{\mathbf{x}}$ submitted by the learner in expectation:

$$\mathbf{w}^* = \operatorname*{argmin}_{\mathbf{w}} \mathbb{E} \left[ \left\| \mathbf{w}^T \mathbf{x}^* - y \right\|^2 \right]$$

As the learner does not have access to the distribution $\mathcal{D}$, but only a sample $S$, the learner can only hope to minimise an empirical version of their loss:

$$\operatorname*{argmin}_{\mathbf{w}} \quad \frac{1}{m} \sum_{i=1}^m \left\| \mathbf{w}^T \mathbf{x}_i^* - y_i \right\|^2$$

$$\text{s.t.} \quad \mathbf{x}_i^* = \operatorname*{argmin}_{\hat{\mathbf{x}}_i} \left\| \mathbf{w}^T \hat{\mathbf{x}}_i - z_i \right\|^2 + \gamma \left\| \mathbf{x}_i - \hat{\mathbf{x}}_i \right\|^2 \quad i \in [m]$$

By constructing a matrix, $X \in \mathbf{R}^{m \times n}$, whose $i$th row corresponds to input example $\mathbf{x}_i$ in the sample $S$, and vectors $\mathbf{y}, \mathbf{z}$, whose $i$th elements correspond to output examples $y_i$ and $z_i$ respectively, we can rewrite this optimisation problem as follows:

$$\operatorname*{argmin}_{\mathbf{w}} \quad ||X^* \mathbf{w} - \mathbf{y}||^2$$

$$\text{s.t.} \quad X^* = \operatorname*{argmin}_{\hat{X}} \left\| \hat{X} \mathbf{w} - \mathbf{z} \right\| + \gamma \left\| \hat{X} - X \right\|_F^2 \tag{1}$$

In the remaining sections, we will develop a practical algorithm which finds global solutions to this optimisation problem.

# 4 Preliminaries

Before we proceed with technical details, we will give a brief overview of how our algorithm is derived, and the optimisation tools used. We begin by reformulating the SPG (1) as a fractional programming problem. We will then claim that such problems can be solved by bisection search by leveraging the classical results of Dinkelbach [13]. Each step of this bisection search requires solving a nonconvex optimisation problem. We will show that this nonconvex optimisation problem can be solved via semidefinite programming through the application of an S-lemma, an equivalence theorem which states when one quadratic inequality is a consequence of a set of other quadratic inequalities.

## 4.1 Dinkelbach's Lemmas for Fractional Programming

Consider the following fractional program:

$$\min_{\mathbf{w}} \quad \frac{N(\mathbf{w})}{D(\mathbf{w})} \quad \text{s.t.} \quad \mathbf{w} \in \mathcal{W} \tag{2}$$

where both $N : \mathbb{R}^n \to \mathbb{R}$ and $D : \mathbb{R}^n \to \mathbb{R}$ are continuous functions and $\mathcal{W}$ is a compact subset of $\mathbb{R}^n$. We can consider the following parameterised family of mathematical programs:

$$\min_{\mathbf{w}} \quad N(\mathbf{w}) - qD(\mathbf{w}) \quad \text{s.t.} \quad \mathbf{w} \in \mathcal{W} \tag{3}$$

where $q \in \mathbb{R}$ is a parameter. In what follows, we will refer to these programs as Dinkelbach Programs. Furthermore, we can define a function $F : \mathbb{R} \to \mathbb{R}$ mapping any $q$ to the solution of its corresponding Dinkelbach Program. Dinkelbach [13] uncovered a number of relationships between $F$ and the fractional program (3), which we shall use in our own analysis:

**Lemma 1** (Dinkelbach [13]). *The function $F(q)$ is continuous in $q$ and is strictly monotonically decreasing. Moreover if $\mathbf{w} \in \mathcal{W}$ and*

$$q = \frac{N(\mathbf{w})}{D(\mathbf{w})}$$

*then $F(q) \leq 0$.*

**Theorem 1** (Theorem 1 from Dinkelbach [13]). *$F(q^*) = 0$ has a unique solution. Furthermore $q^*$ is the solution to the Fractional Program (3) if and only if $F(q^*) = 0$.*

In particular, in Section 6, we shall exploit Theorem 1 to circumvent the nonconvexities present in the fractional formulation of our SPG.

## 4.2 S-lemma with Equality

Consider a pair of quadratic functions:

$$f(\mathbf{w}) = \mathbf{w}^T A \mathbf{w} + 2\mathbf{a}^T \mathbf{w} + c \geq 0$$
$$h(\mathbf{w}) = \mathbf{w}^T B \mathbf{w} + 2\mathbf{b}^T \mathbf{w} + d = 0$$

where $A, B \in \mathbb{R}^{n \times n}$ are symmetric matrices, $\mathbf{a}, \mathbf{b} \in \mathbb{R}^n$ and $c, d \in \mathbb{R}$. An S-lemma with equality is a theorem which specifies the conditions under which the following two statements are equivalent:

(i) $h(\mathbf{w}) = 0 \implies f(\mathbf{w}) \geq 0 \quad \forall \mathbf{w} \in \mathbb{R}^n$

(ii) There exists a number $\lambda$ such that

$$f(\mathbf{w}) + \lambda h(\mathbf{w}) \geq 0 \quad \forall \mathbf{w} \in \mathbb{R}^n$$

Nearly all S-lemmas with equality assume that the dual Slater condition holds:

**Assumption 1** (Dual Slater Condition). *There exists $\mathbf{w}_1 \in \mathbb{R}^n$ and $\mathbf{w}_2 \in \mathbb{R}^n$ such that $h(\mathbf{w}_1) < 0$ and $h(\mathbf{w}_2) > 0$.*

The main application of S-lemmas is their use in transforming quadratically constrained quadratic programs with a single constraint (QC1QPs) into convex semidefinite programs (SDPs), which can be solved efficiently via interior point methods [3]. After we derive our bisection search algorithm, we will see that each step of bisection search requires solving a QC1QP. We will make use of the following S-lemma with equality, proved by Xia et al. [25], to reformulate the QC1QP at each iteration into an SDP.

**Theorem 2** (Theorem 3 from Xia et al. [25]). *Assume that assumption 1 holds. In addition, assume that $B \neq 0$. Then statements (i) and (ii) are equivalent.*

# 5 Problem Reformulation

Before we can apply the theorems described in the previous section, we first need to reformulate our optimisation problem into a more amenable form. We start by noting that the lower level optimisation problem in SPG (1) is in fact convex and therefore its solution can be found by finding a point at which the gradient with respect to $\hat{X}$ is equal to zero. This yields a closed form solution for $X^*$ which can be used to rewrite the constraint in problem (1):

$$\operatorname*{argmin}_{\mathbf{w}} \quad ||X^*\mathbf{w} - \mathbf{y}||^2$$
$$\text{s.t.} \quad X^* = (\mathbf{z}\mathbf{w}^T + \gamma X)(\mathbf{w}\mathbf{w}^T + \gamma I)^{-1}$$

Substituting the right hand side of the constraint directly into the objective allows us get rid of constraints entirely. Moreover, applying the Sherman-Morrison formula [3] to the matrix $\mathbf{w}\mathbf{w}^T + \gamma I$ allows us to derive an equivalent nonlinear least squares problem:

$$\operatorname*{argmin}_{\mathbf{w}} \quad \left|\left| \frac{\frac{1}{\gamma}\mathbf{z}\mathbf{w}^T\mathbf{w} + X\mathbf{w}}{1 + \frac{1}{\gamma}\mathbf{w}^T\mathbf{w}} - \mathbf{y} \right|\right|^2 \tag{4}$$

Intuitively, we can see that the learner is minimising the error of a weighted average between the target labels $\mathbf{z}$ of the adversary and the predictions their chosen model would make on the true data matrix $X$. As the learner chooses a model bigger in Euclidean norm, predictions will be slowly pushed towards the desired labels of the adversary. This makes sense, as the adversary is constrained in Frobenius norm, so by making the weight vector bigger in $\ell_2$-norm, the learner makes it cheaper for the adversary to manipulate. We also see that, as $\gamma$ grows, the slower the weight on the target labels of the adversary grows as the $\ell_2$-norm of the weight vector increases.

By moving $\mathbf{y}$ into the fraction we can express problem (4) as the following fractional programming problem:

$$\operatorname*{argmin}_{\mathbf{w}} \quad \frac{\left|\left| \frac{1}{\gamma}\mathbf{z}\mathbf{w}^T\mathbf{w} + X\mathbf{w} - \mathbf{y} - \frac{1}{\gamma}\mathbf{w}^T\mathbf{w}\mathbf{y} \right|\right|^2}{(1 + \frac{1}{\gamma}\mathbf{w}^T\mathbf{w})^2}$$

To get rid of high degree terms, we introduce a new variable $\alpha \in \mathbb{R}$ and set it equal to $\mathbf{w}^T\mathbf{w}$:

$$\operatorname*{argmin}_{\mathbf{w},\alpha} \quad \frac{\left|\left| \frac{\alpha}{\gamma}\mathbf{z} + X\mathbf{w} - \mathbf{y} - \frac{\alpha}{\gamma}\mathbf{y} \right|\right|^2}{(1 + \frac{\alpha}{\gamma})^2} \quad \text{s.t.} \quad \alpha = \mathbf{w}^T\mathbf{w} \tag{5}$$

We are now left with a quadratic fractional programming problem with a nonconvex feasible set. At this point, we are ready to begin applying the techniques discussed in Section 4 in order to develop a practical algorithm.

# 6 Solving the Fractional Program

We will now apply the theorems and lemmas of Dinkelbach to construct an algorithm for solving problem (5). To begin, we first consider the Dinkelbach problems associated with problem (5):

$$F(q) = \min_{\mathbf{w},\alpha} \quad \left|\left| \frac{\alpha}{\gamma}\mathbf{z} + X\mathbf{w} - \mathbf{y} - \frac{\alpha}{\gamma}\mathbf{y} \right|\right|^2 - q\left(1 + \frac{\alpha}{\gamma}\right)^2$$
$$\text{s.t.} \quad \alpha = \mathbf{w}^T\mathbf{w}$$

By Theorem 1, we know that finding a $q^* \in \mathbb{R}$ such that $F(q^*) = 0$, implies that the linear predictor $\mathbf{w}^*$ corresponding to $F(q^*)$ is a global solution to problem (5), and therefore, the original SPG. We also know from Lemma 1 that $F$ is a concave monotonically decreasing continuous function. As a result, given $q_1, q_2 \in \mathbb{R}$ for which $F(q_1) \leq 0$ and $F(q_2) \geq 0$ we can employ bisection search to find $q^*$. This procedure is described formally by Algorithm 1.

Given the description of our algorithm, we now show that it indeed converges linearly to a globally optimal solution. In particular, we have the following:

**Algorithm 1** Bisection search for $q^*$

---

**Input:** data matrix $X$, learner's labels $\mathbf{y}$, data provider's labels $\mathbf{z}$, tolerance $\epsilon$
Initialize $q_1 = 0$
Initialize $q_2 = \mathbf{y}^T \mathbf{y}$
**repeat**
    $q = (q_1 + q_2)/2$
    **if** $F(q) \geq 0$ **then**
        $q_1 = q$
    **else**
        $q_2 = q$
    **end if**
**until** $q_2 - q_1 \leq \epsilon$
**return** $q_2$

---

**Theorem 3.** *Algorithm 1 takes at most $\log_2(\frac{2\mathbf{y}^T\mathbf{y}}{\epsilon})$ iterations to return a $q \in \mathbb{R}$ such that $q - q^* \leq \epsilon$*

*Proof.* For bisection search to converge to $q^*$ such that $F(q^*) = 0$ we require two points. One point, $q_1 \in \mathbb{R}$, for which $F(q_1) \geq 0$, and another point, $q_2 \in \mathbb{R}$, for which $F(q_2) \leq 0$. We begin by claiming that setting $a = 0$ and $b = \mathbf{y}^T \mathbf{y}$ satisfies these conditions.

Finding a $q_2$ such that $F(q_2) \leq 0$ is simple. Lemma 1 tells us that we can choose any feasible point for problem 5) and simply set $q_2$ to the value of the objective at the chosen point. Thus we choose the zero vector which leads to an objective value of $\mathbf{y}^T \mathbf{y}$.

To find a $q_2$ such that $F(q) \geq 0$, we employ both Lemma 1 and Theorem 1. Since $F(q^*) = 0$ and $F$ is strictly monotonically decreasing, any value which lower bounds $q^*$ will map to a nonnegative value when passed to $F$. Since the objective of problem (5) is always nonnegative, we can select $q_2 = 0$ as a lower bound.

Since $F(q_1)$ is nonnegative, $F(q_2)$ is nonpositive and $F$ is continuous, $q^*$ must lie in the interval $[q_1, q_2]$ by the intermediate value theorem. Thus $q = \frac{q_1 + q_2}{2}$, the mid-point of $[q_1, q_2]$, is at most $|\frac{q_1 - q_2}{2}|$ in distance from $q^*$. Note that, in Algorithm 1, $q_1$ is initialised at 0 and $q_2$ is initialised at $\mathbf{y}^T \mathbf{y}$. Thus, initially $|q_1 - q_2| = \mathbf{y}^T \mathbf{y}$.

At each iteration, the interval $[q_1, q_2]$ is updated and halved in length. Therefore, after $\log_2(\frac{2\mathbf{y}^T\mathbf{y}}{\epsilon})$ iterations $|q - q^*| \leq \frac{\epsilon}{2}$. After the same number of iterations $q_2 - q = \frac{\epsilon}{2}$. Therefore, $q_2 - q^* \leq \epsilon$. As $q_2$ is returned by Algorithm 1, we have proved the result. $\square$

Theorem 3 essentially states that, if it is possible to evaluate $F(q)$, then it is possible to converge linearly to a global solution. In contrast, existing methods can only guarantee convergence to stationary points which satisfy the Karush-Kuhn-Tucker (KKT) Conditions. In addition, Theorem 3 indicates that this subclass of bilevel optimisation problems are easy in comparison to general bilevel optimisation problems as long as $F(q)$ can be evaluated cheaply.

A disadvantage of Algorithm 1 is that each iteration of the algorithm requires an evaluation of $F$, where $F$ itself is in general a nonconvex optimisation problem. Fortunately, $F(q)$ is a QC1QP. We can see this by rewriting $F(q)$ as follows:

$$\begin{aligned} \min_{\hat{\mathbf{w}}} \quad & f(\hat{\mathbf{w}}) = \hat{\mathbf{w}}^T A \hat{\mathbf{w}} + 2\mathbf{a}^T \hat{\mathbf{w}} + c \\ \text{s.t.} \quad & h(\hat{\mathbf{w}}) = \hat{\mathbf{w}}^T B \hat{\mathbf{w}} + 2\mathbf{b}^T \hat{\mathbf{w}} = 0 \end{aligned} \tag{6}$$

where

$$\hat{\mathbf{w}} = \begin{bmatrix} \mathbf{w} \\ \alpha \end{bmatrix}, \quad \mathbf{b} = \begin{bmatrix} \mathbf{0} \\ 1 \end{bmatrix}, \quad \mathbf{a} = \begin{bmatrix} -X^T \mathbf{y} \\ -\frac{1}{\gamma}(\mathbf{z} - \mathbf{y})^T \mathbf{y} - \frac{q}{\gamma} \end{bmatrix}, \quad B = \begin{bmatrix} -I & \mathbf{0} \\ \mathbf{0}^T & 0 \end{bmatrix}, \quad c = \mathbf{y}^T \mathbf{y} - q$$

and

$$A = \begin{bmatrix} X^T X & \frac{1}{\gamma} X^T (\mathbf{z} - \mathbf{y}) \\ \frac{1}{\gamma} X^T (\mathbf{z} - \mathbf{y}) & \frac{1}{\gamma^2}(\mathbf{z} - \mathbf{y})^T (\mathbf{z} - \mathbf{y}) - \frac{q}{\gamma^2} \end{bmatrix}$$

We can now proceed with an S-procedure (the application of an S-lemma), in order to reformulate problem (6) as an SDP, which is convex and can be solved via interior point methods. First, we rewrite problem (6) as the following equivalent problem:

$$\max_{\tau} \quad \tau \quad \text{s.t.} \quad h(\hat{\mathbf{w}}) = 0 \implies f(\hat{\mathbf{w}}) - \tau \geq 0 \quad \forall \hat{\mathbf{w}} \in \mathbb{R}^{n+1}$$

Since $B \neq 0$ we can apply Theorem 2 to rewrite the constraint by introducing a new variable $\lambda \in \mathbb{R}$:

$$\max_{\tau, \lambda} \quad \tau \quad \text{s.t.} \quad f(\hat{\mathbf{w}}) - \tau + \lambda h(\hat{\mathbf{w}}) \geq 0 \quad \forall \hat{\mathbf{w}} \in \mathbb{R}^{n+1} \tag{7}$$

Furthermore, note that the constraint is equivalent to the following inequality:

$$\begin{bmatrix} \hat{\mathbf{w}} \\ 1 \end{bmatrix}^T \begin{bmatrix} A + \lambda B & \mathbf{a} + \lambda \mathbf{b} \\ \mathbf{a}^T + \lambda \mathbf{b}^T & c - \tau \end{bmatrix} \begin{bmatrix} \hat{\mathbf{w}} \\ 1 \end{bmatrix} \geq 0 \quad \forall \hat{\mathbf{w}} \in \mathbb{R}^{n+1}$$

In what follows, we will refer to the matrix in the above constraint by $M$. It is easy to see that the constraint above is equivalent to a positive semidefinite constraint on $M$. For the sake of contradiction, suppose that $\exists \mathbf{x} \in \mathbb{R}^{n+2}$ such that $\mathbf{x}^T M \mathbf{x} < 0$, but the constraint above holds. If the final coordinate of $\mathbf{x}$ is nonzero, we can simply rescale $\mathbf{x}$ by this coordinate to achieve a contradiction. If the final coordinate of $\mathbf{x}$ is nonzero, then we can use the continuity of quadratic forms to argue that there must exist a $\bar{\mathbf{x}}$ whose final coordinate is not equal to zero such that $\bar{\mathbf{x}}^T M \bar{\mathbf{x}} < 0$, bringing us back to the nonzero case. Thus, we can replace the constraint in (7) as follows:

$$\max_{\tau, \lambda} \quad \tau \quad \text{s.t.} \quad \begin{bmatrix} A + \lambda B & \mathbf{a} + \lambda \mathbf{b} \\ \mathbf{a}^T + \lambda \mathbf{b}^T & c - \tau \end{bmatrix} \succeq 0 \tag{8}$$

Thus, evaluating $F(q)$ corresponds to solving the SDP (8), which can be solved efficiently using interior point methods. Assuming that strong conic duality holds, then a value for $\hat{\mathbf{w}}$ which attains $F(q)$ can be found by taking a rank-1 decomposition of dual variables. A description of the dual program can be found in the supplementary materials.

Moreover, as each SDP has rank-1 solution, we can apply standard low-rank approximation schemes for SDPs in order to cope with data of high dimension [5, 6]. In addition, a variety of first-order methods tailored to SDPs can be applied in scenarios where evaluating the Hessian is prohibitively expensive [22].

## 7 Empirical Evaluation

Given the theoretical performance analysis of our proposed algorithm, we now demonstrate that it is significantly more accurate in practice compared to state of the art approaches. To this end, we evaluate Algorithm 1 on two real world datasets and compare it against both ridge regression, which is the optimal approach under the assumption that data providers are completely adversarial, and to the single level nonconvex reformulation of the SPG originally proposed by Brückner and Scheffer [4]. In what follows, we discuss the results with respect to one of these datasets, the medical personal costs dataset [9], and defer results regarding the second dataset, the red wine dataset [11], to the supplementary material.

### 7.1 Medical Personal Costs Dataset

The medical personal costs dataset consists of 1338 instances each with 7 features [9]. Each feature details information regarding an individual. Some are continuous, such as the individual's age and body mass index, while others are categorical, such as region and smoking status. So that we can perform linear regression on all features, categorical features are transformed into one-hot vectors, increasing the total number of features from 7 to 13. The response variable is the individual's medical costs billed by their health insurance.

We consider a scenario in which insurers would like to predict the medical costs of a new customer in order to provide a reasonable insurance quote. We assume that individuals may provide fake data in the hope of receiving a lower quote. Similarly to the experimental design of Tong et al. [24], we define each data provider's desired outcome as $\mathbf{z}_i = \mathbf{y}_i + \delta_i$, where $\delta_i$ denotes the change in medical billing that each individual is striving for. In particular, we consider two types of provider, $\mathcal{A}_{\text{modest}}$, who wishes to reduce their predicted medical billing by \$100 ($\delta_{\text{modest}} = -100.0$), and $\mathcal{A}_{\text{severe}}$, who

wishes to reduce their predicted medical billing by \$300 ($\delta_{\text{severe}} = -300.0$). Since medical charges range from \$1000 to \$63,000, we numerically scale the data labels by dividing them by 100 before passing them to each algorithm.

In order to evaluate Algorithm 1, we perform 10-fold cross validation and compare its performance to ridge regression for $\gamma \in [1 \times 10^{-5}, 1]$. For each value of $\gamma$, we compute the regularisation parameter for ridge regression via grid search on 8 logarithmically spaced points in the interval $[1 \times 10^{-5}, 1000]$ during cross validation. For ridge regression, and the SDPs in Algorithm 1, we use the SDPT3 solver [23] to find global solutions. We also compare Algorithm 1 to the nonconvex single level reformulation of the SPG originally proposed by Brückner and Scheffer [4]. We employ the interior point method from the MATLAB optimisation toolbox to find a stationary point for this problem reformulation. The same error tolerances are used for both Algorithm 1 and the interior point method we use to solve noncovex problem reformulation proposed by Brückner and Scheffer [4]. The results of this comparison are given in Figure 1.

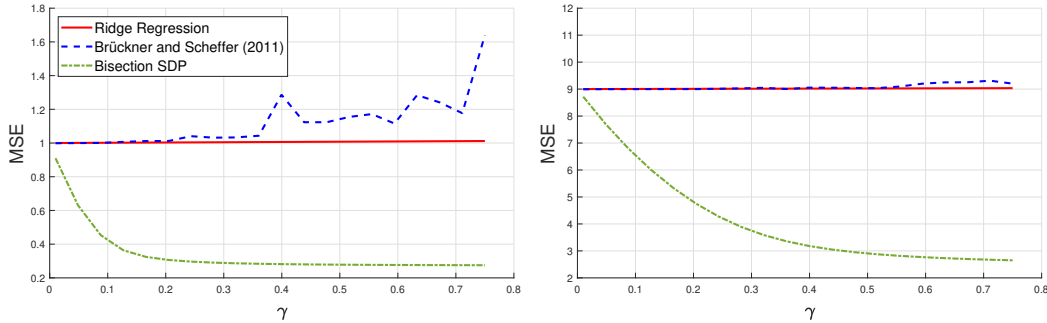

Figure 1: A performance comparison between different algorithms ran on the medical personal costs dataset. The left plot compares the average MSE of each algorithm during 10-fold cross validation where data was generated by $\mathcal{A}_{\text{modest}}$, whilst the right plot shows the average MSE where data is generated by $\mathcal{A}_{\text{severe}}$

## 7.2 Results Analysis

Figure 1 shows the average mean squared error (MSE) achieved by each algorithm on the medical personal costs dataset. Firstly, observe that Algorithm 1 outperforms both ridge regression and the nonconvex problem reformulation for every value of $\gamma$. Also note that the MSE of Algorithm 1 is very stable, whilst the interior point solution seems to behave erratically for higher values of $\gamma$.

For values of $\gamma > 0.4$, we observe that, for modest data providers, our algorithm is at least \$45 more accurate than ridge regression on average. For severe data providers, we observe an even greater difference. For example, when $\gamma > 0.5$, the predictions made by our algorithm are at least \$120 more accurate than ridge regression on average. As one would expect, the benefits of explicitly modelling the goals of data providers becomes more beneficial as $\gamma$ increases, as the data provider's capability for manipulation becomes more limited. In the supplementary material we have included numerical experiments conducted on another real world dataset (the red wine dataset [11]), which yield similar results.

## 8 Conclusions and Future Work

In this paper, we studied a particular subclass of SPGs, in which data providers have their own goal of manipulating the learner into selecting a given outcome. We showed that this bilevel optimisation problem is equivalent to solving a series of QC1QPs. Using a combination of fractional programming and the S-lemma, we derived an algorithm which linearly converges to a global solution. Lastly, we evaluated our derived algorithm on real world datasets and showed that our algorithm outperforms existing methods in terms of mean squared error.

One disadvantage of our approach is that we require a verified training dataset which details the motivations of data providers. In many cases, it may not be possible to know the ambitions of data providers apriori. In future work, we will consider an online version of this problem, in which the learner must choose a learning model based on the past data it has observed and its current beliefs regarding the motivations of the data provider. We also assumed that data providers were punished for making manipulations in the $\ell_2$-norm. It would be interesting to explore what happens when data providers are constrained differently, and whether algorithms of similar structure exist in such cases.

## 9 Broader Impact

The manipulation and fairness of algorithms form a significant barrier to practical application of theoretically effective machine learning algorithms in many real world use cases. With this work, we have attempted to address the important problem of data manipulation, which has many societal consequences. Data manipulation is one of many ways in which an individual can "game the system" in order to secure beneficial outcomes for themselves to the detriment of others. Thus, reducing the potential benefits of data manipulation is of worthwhile consideration and focus. Whilst this paper is primarily of theoretical focus, we hope that our work will form a contributing step towards safe, fair, and effective application of machine learning algorithms in more practical settings.

## Acknowledgments and Disclosure of Funding

This work was supported by the UK Engineering and Physical Sciences Research Council (EPSRC) Doctoral Training Partnership grant.

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
