[Supplementary Material · Neurips_SPG_camera_supp.pdf]

# A The Dual Problem

In this section, we describe the dual of the SDP that we solve at each iteration of Algorithm 1. This dual can be used to obtain a linear predictor at every step of the algorithm. Recall the SDP which is solved at each time step of Algorithm 1:

$$\max_{\tau,\lambda} \quad \tau \quad \text{s.t.} \quad \begin{bmatrix} A + \lambda B & \mathbf{a} + \lambda \mathbf{b} \\ \mathbf{a}^T + \lambda \mathbf{b}^T & c - \tau \end{bmatrix} \succeq 0 \tag{9}$$

We can rewrite this SDP as follows:

$$\max_{\tau,\lambda} \quad \begin{bmatrix} 1 \\ 0 \end{bmatrix}^T \begin{bmatrix} \tau \\ \lambda \end{bmatrix} \quad \text{s.t.} \quad \lambda \begin{bmatrix} -B & -\mathbf{b} \\ -\mathbf{b}^T & 0 \end{bmatrix} + \tau \begin{bmatrix} \mathbf{0} & \mathbf{0} \\ \mathbf{0} & 1 \end{bmatrix} \preceq \begin{bmatrix} A & \mathbf{a} \\ \mathbf{a}^T & c \end{bmatrix}$$

Taking the dual yields the following SDP:

$$\min_{W \in \mathbb{S}^{n+2}} \quad \begin{bmatrix} A & \mathbf{a} \\ \mathbf{a}^T & c \end{bmatrix} \cdot W \quad \text{s.t.} \quad \begin{bmatrix} B & \mathbf{b} \\ \mathbf{b}^T & 0 \end{bmatrix} \cdot W = 0, \quad \begin{bmatrix} \mathbf{0} & \mathbf{0} \\ \mathbf{0} & 1 \end{bmatrix} \cdot W = 1, \quad W \succeq 0 \tag{10}$$

Taking an appropriate rank-1 decomposition of the optimal solution to (10) yields a vector containing a linear predictor and its squared euclidean norm. As a direct consequence of the results in Section 6, this linear predictor is optimal for the Dinkelbach program associated with the primal SDP (9).

# B Extensions to Kernel Methods

In this section we briefly describe a version of Algorithm 1 based on kernel methods. In some instances, we may wish to apply a high dimensional feature mapping, $\phi$, to data points before making a prediction. Typically, it is assumed that the feature mapping $\phi$ maps each element of $\mathcal{X}$ to an element of a reproducing Hilbert kernel space (RKHS) $\mathcal{F}$, with corresponding kernel function $k : \mathcal{X} \times \mathcal{X} \to \mathcal{R}$. In the standard least squares linear regression setting, where data is sampled cleanly without manipulation, this leaves us with the following optimisation problem:

$$\operatorname*{argmin}_{\mathbf{w}} \sum_{i=1}^{m} (\mathbf{w}^T \phi(\mathbf{x}_i) - y_i)^2$$

By leveraging the representer theorem [20], it can be shown that there exists an optimal solution $\mathbf{w}^* \in \mathcal{F}$ with the following form:

$$\mathbf{w}^* = \sum_{i=1}^{m} \beta_i \phi(\mathbf{x}_i) \tag{11}$$

where $\beta_i \in \mathbb{R}$ for all $i \in [m]$. Moreover, the coefficients $\beta_i$ can be characterised by the Gram matrix $K \in \mathbb{R}^{m \times m}$, where $K_{ij} = k(\mathbf{x}_i, \mathbf{x}_j)$. In contrast, under the assumption that the feature map $\phi$ is surjective, for the SPG problem setting discussed in this paper, we are left with the following optimisation problem:

$$\operatorname*{argmin}_{\mathbf{w}} \quad \sum_{i=1}^{m} \left( \frac{\frac{1}{\gamma} z_i \mathbf{w}^T \mathbf{w} + \mathbf{w}^T \phi(\mathbf{x}_i)}{1 + \frac{1}{\gamma} \mathbf{w}^T \mathbf{w}} - \mathbf{y} \right)^2$$

In this case, the representer theorem cannot be applied, as the prediction made by a given predictor $\mathbf{w}$ depends on its inner product with each mapped data point *and* its own norm, rather than just the former. However, if we optimise over predictors of the form described in equation (11), then we can obtain a new version of Algorithm 1 which returns the optimal vector of coefficients, $\beta = (\beta_1, \ldots \beta_m)^T$, wherein the following terms are redefined:

$$\mathbf{b} = \begin{bmatrix} \mathbf{0} \\ 1 \end{bmatrix}, \quad \mathbf{a} = \begin{bmatrix} -K^T \mathbf{y} \\ -\frac{1}{\gamma}(\mathbf{z} - \mathbf{y})^T \mathbf{y} - \frac{q}{\gamma} \end{bmatrix}, \quad B = \begin{bmatrix} -I & \mathbf{0} \\ \mathbf{0}^T & 0 \end{bmatrix}, \quad c = \mathbf{y}^T \mathbf{y} - q$$

and

$$A = \begin{bmatrix} K^T K & \frac{1}{\gamma} K(\mathbf{z} - \mathbf{y}) \\ \frac{1}{\gamma} K^T(\mathbf{z} - \mathbf{y}) & \frac{1}{\gamma^2}(\mathbf{z} - \mathbf{y})^T(\mathbf{z} - \mathbf{y}) - \frac{q}{\gamma^2} \end{bmatrix}$$

Whilst we cannot guarantee that the predictor output by this algorithm will be optimal, we can guarantee that the predictor is better than any other predictor in the span of mapped training data points, which by the representer theorem, contains the optimal predictor for the classical clean data setting.

## C    Additional Experiments

In this section, we compare Algorithm 1 to ridge regression and the nonconvex relaxation of Brückner and Scheffer [4] using the red wine dataset [11]. The red wine dataset [11] contains 1599 instances each with 11 features. Each feature is a sensory or physiochemical measurement for wine. The response variable is a wine rating out of 10 points, where 10 is the best rating possible.

We place ourselves in the position of wine producers, who may wish to increase the rating of their wine by submitting fake input data. We assume that a wine producer is happy with the rating for their wine if it is greater than or equal to a threshold, $t \in [0, 10]$. Thus, if the true output label associated with a wine is greater than or equal to the threshold, then the target label of the data provider is identical to the true output label. Otherwise, the target output label of the data provider is set to $t$. Formally:

$$z_i = \max\{y_i, t\} \tag{12}$$

Similarly to experiments conducted on the medical personals costs dataset, we perform 10-fold cross validation on the dataset and average the MSE of each approach over each fold for $\gamma \in [1 \times 10^{-5}, 0.2]$. As before, a ridge regression hyperparameter is selected for each $\gamma$ by grid search on 8 logarithmically spaced points in the interval $[1 \times 10^{-5}, 1000]$. We define two different data providers, each with a different threshold: a modest data provider with $t_{\mathrm{modest}} = 6$, and a severe data provider with $t_{\mathrm{severe}} = 8$. The results of the experiments are shown in Figure 2. As in the case of the medical personal costs daataset, Algorithm 1 outperforms other approaches for all values of $\gamma$. For values of $\gamma > 0.1$, we observe that, for modest data providers, our algorithm is at least 0.1 points more accurate than ridge regression on average. For severe data providers, we observe an even greater difference. Meanwhile, for severe data providers, Algorithm 1 is more than an entire point more accurate than ridge regression on average.

## D    Run Time Comparison

Lastly, we compare the run time of our algorithm to that of the interior point method for finding local solutions to the nonconvex relaxation of Brückner and Scheffer [4]. For these experiments, we fix $\gamma = 0.5$. Using the medical personal costs dataset, we run our algorithm and the interior point method on training sets of varying sizes. For each training set size, we create 10 training sets by sampling uniformly from the entire dataset. This experiment was on run on an AMD Ryzen 1600 3.20GHz six-core processor on a single thread of execution. The error tolerances of the interior point method and Algorithm 1 are set to $1 \times 10^{-2}$.

Figure 3 shows the mean run time of each algorithm for different problems sizes, with each error bar representing a 95% confidence interval according to the student t-distribution. Note that, in all cases, the interior point method quickly grows in run time as the scale of the problem increases, whilst Algorithm 1 has a similar running time for all problem scales tested. Whilst the interior point approach is faster on smaller problem instances, it quickly becomes apparent that Algorithm 1 is faster on larger instances. Also note that the run time of Algorithm 1 is far more consistent, and has far less variance, especially as the problem scale grows.

It is worth noting that, for every problem scale, we use the same initial upper bound for $q$ in our bisection search. More specifically, we take the entire medical personal costs dataset and upper bound $q$ using the inner product of the training labels. As a result, the number of SDPs that need to be solved to achieve the same error tolerance for different problem scales is roughly equivalent. This partly explains why the performance of the bisection method stays relatively constant across different problem scales in our experiments.

Figure 2: A performance comparison between different algorithms run on the red wine dataset in which the target labels of each data provider are given by equation (12). The left plot corresponds to experiments run with $t_{\text{modest}}$, whilst the right plot corresponds experiments run with $t_{\text{severe}}$.

Figure 3: A run time comparison between the interior point method approach and Algorithm 1 using the medical personal costs dataset. The plot on the left corresponds experiments run with $\mathcal{A}_{\text{modest}}$, whilst the right plot corresponds to experiments run with $\mathcal{A}_{\text{severe}}$.