[Reviews · NeurIPS 2020]

Review 1

Summary and Contributions: This paper considers a problem of linear regression where some of your inputs are strategically corrupted. In particular, the algorithm is given some vector data x, which it uses to approximate y by w.x for some learned value w. However, an adversary can corrupt the value of x to x^ in an attempt to make the predicted value closer to z. Given a training set of triples (x,y,z) the objective is to learn a w that does as well as possible despite these corruptions. Formally, the adversary sends the algorithm x^ that minimizes |w.x^-z|^2+gamma|x-x^|^2. In particular, they try to minimize the error between the prediction and x^ without making x^ too far from z. The algorithm tries to pick a w so that the average value of |w.x^-y|^2 is minimized.

Strengths: This is a natural problem to consider and the solution involves some clever manipulations and optimization algorithms.

Weaknesses: The results are very specific to the particular model, which although natural, seems narrow in potential application.

Correctness: The results appear to be correct.

Clarity: The paper is reasonably easy to understand.

Relation to Prior Work: It is not clear from the paper how their model differs from previously considered ones.

Reproducibility: Yes

Additional Feedback:


Review 2

Summary and Contributions: This paper describes a Stackelberg prediction model for offline least squares regression in the case where data comes from multiple data providers, each of which has a goal for the learner's outcome on its provided data. The authors reformulate the considered problem as a fractional programming problem, and then provide a fast algorithm for solving it based on the results of Dinkelbach. The paper then gives an empirical evaluation on toy data.

Strengths: I am not an expert in non-convex optimization/SPGs. Therefore I cannot comment on the strength of this paper with respect to its significance/novelty or technical contributions.

Weaknesses: Motivation of setting: In the proposed model, the learner tries to learn a linear predictor that does well on the manipulated data, while the adversary manipulates points with the goal of having the learner minimize an objective on the *unmanipulated data*. What is the motivation for this mismatch? (E.g. why would the learner ever evaluate on both the manipulated and unmanipulated data in practice?)

Correctness: Which line in Figure 1 corresponds to the previous work of Bruckner et al? I am guessing it is the "interior point" line --- this seems suspiciously bad. Is this a strong baseline / was the baseline correctly executed with proper hyper-parameter choice etc? Is there any way to contextualize the MSE in Figure 1? Currently it is unclear how to relate the MSE to the solved problem in a meaningful way.

Clarity: The paper is well written and was relatively easy to follow, especially as an outsider in the field. Grammar: line 89, 93-94, end of Figure 1 caption

Relation to Prior Work: The authors give a detailed related work section that made me feel like I had a good overview of the considered problem and related approaches.

Reproducibility: Yes

Additional Feedback: Should the \hat{X} in the first term of the constraint in (1) be X? (Otherwise (1) is not the same optimization problem as the non-vectorized version in lines 115-116). ----- Post response feedback: The response addressed my concerns over the setup and the experimental results, but after reviewing the other reviewer responses in combination with the response I still am not sure of the significance of the results.


Review 3

Summary and Contributions: In this paper, the authors study the possibility of achieving the global optimality of a special non-convex problem. The problem is about a self-interested agent who could manipulate the data sampling process at a cost so that he cannot arbitrarily modify the data. The authors showed that under the case of a linear, MSE-evaluated learner, such a non-convex problem could be efficiently solved with its global optimality.

Strengths: 1. A significant step towards solving a class of practical and impactful non-convex problems. 2. Remarkable consistency of its theory and practice. 3. Motivating towards a provable solution of global optimality of a class of problems in non-uniform data sampling scenarios.

Weaknesses: The experiment is impressive, but it could be better if we had more of it. Also, it seems to me that the analysis cannot be directly extended to more complicated cases beyond linear regression. I was wondering if the author could revisit the analysis and come up with a relaxed, general conclusion with less assumption on the value function y = f(x).

Correctness: I am aware of the correctness of the paper.

Clarity: Yes.

Relation to Prior Work: Yes.

Reproducibility: Yes

Additional Feedback: I don't feel I have understood every bit of the paper according to previous discussion with other reviewers and the rebuttal. I tend to give a score of 7 with low confidence 2.

[Author Response · NeurIPS 2020]

Firstly, we thank the reviewers for their valuable comments. We will address the comments of each reviewer in what follows.

**R1, R4**: *The results are very specific to the particular model*: Indeed it is the case that our theoretical results assume that data providers are constrained in $\ell_2$-norm, and that both the learner and the data providers are interested in solving linear least squares problems. However, it is fairly easy to see that our theoretical results generalise to kernel ridge regression, and thus, our theoretical results hold for a far wider array of learning models than linear predictors. In addition, we believe that our work is an important first step in relaxing the overly pessimistic assumptions of adversarial machine learning. Whilst it is not reasonable in practice to assume that data is sampled i.i.d. from the distribution of interest, neither is it reasonable to assume that data is provided with the sole intention of hindering learning. By taking the incentives of data providers directly into account during the optimisation process, we hope to better reflect the reality of sampled data in practice. As previously stated, we believe our work forms a first step in achieving this goal.

**R1**: *It is not clear from the paper how their model differs from previously considered ones*: In this paper, we study a specific subclass of SPGs in which both the learner and data providers are interested in solving least squares linear regression problems with their own data labels. Whilst Brückner and Scheffer (2011) give an algorithm which converges to local optima for general SPGs, we give an algorithm for this specific subclass of SPGs which converges to **globally optimal solutions**. Note that SPGs are bilevel optimisation problems, which are, in general, NP-hard. With this in mind, we believe our results are novel and significant, as we have provided a practically efficient algorithm for a large subset of bilevel optimisation problems. Algorithms for specific subclasses of SPGs have been considered before, but to the best of our knowledge, our paper is the first to consider SPGs for linear least squares regression. Closely related to our work is the problem of robust regression in which the challenge is to choose a model which performs well in the presence of worst-case noise. The subclass of SPGs we consider allow us to model data providers with more nuanced motivations, which we believe are more likely to arise in practice. We will highlight these differences from previous work in future versions of the paper.

**R2**: *Why would the learner ever evaluate on both the manipulated and unmanipulated data in practice?*: We believe that our work has applications whenever a learner has access to a reliable, but costly, verification process. In practice, many machine learning models make decisions which affect those who provide data. Thus, data providers may manipulate the data they submit in order to obtain a preferred outcome. In cases in which the learner has access to a verification process, the learner can recover the originally sampled data as well as the goals of the data providers. Unfortunately, this verification process may be too expensive to use on every single data point. However, the learner may be able to verify a sample which can be used to learn the motivations of data providers and select a model which anticipates the manipulations that data providers are likely to make. We believe that our theoretical model captures this dynamic. Note that using verified data points to improve learning has been applied extensively in adversarial machine learning contexts (for examples, see Charikar *et al.* (2017) and Raghavendra and Yau (2020)). One practical example of such a setting is insurance fraud. An insurance company may gather information from a customer to better evaluate potential risk. However, a smart customer, who knows the information they provide could be used to decide the cost of their insurance, may lie when submitting their information. Whilst insurance companies can often verify information regarding their customers, verification is often expensive, requiring a significant deployment of staff and/or resources.

**R2**: *I am guessing it is the "interior point" line - this seems suspiciously bad. Is this a strong baseline / was the baseline correctly executed with proper hyper-parameter choice?*: The reviewer is correct in their assumption that the interior point line in figure 1 corresponds to the method of Brückner and Scheffer. We shall update the legend of figure 1 to be more clear. This approach involves reformulating SPGs into a single-level optimisation problem which can be solved via conventional optimisation techniques. In their work, Brückner and Scheffer highlight a number of SPGs for which this problem reformulation is simple and can be easily solved. For the family of SPGs we consider, the problem reformulation is nontrivial and nonconvex. The error tolerances used for both our algorithm and the interior point method we use to solve this problem reformulation are the same. We believe that the poor performance of the interior point method for higher values of $\gamma$ reflects ill-conditioning issues present in the nonconvex objective, but we cannot confirm this.

**R2**: *Is there any way to contextualize the MSE in Figure 1?*: A data label in the Medical Personal Costs dataset corresponds to the medical charges for a given individual. Since these charges range from $1000 to $63,000, we numerically scale the data labels by dividing them by 100 before passing them to each algorithm. For values of $\gamma > 0.4$, we observe that, for modest data providers, our algorithm is at least more accurate by $45 on average. For severe data providers, we observe an even greater difference. For example, when $\gamma > 0.5$, the predictions made by our algorithm are at least $120 more accurate on average. Considering the modest data provider only sought to reduce their charges by $100, and the severe data provider only sought to reduce their charges by $300, we believe these differences are fairly significant. We shall provide more contextualisation for our empirical experiments in future versions of the paper.

[Meta-Review · NeurIPS 2020]

This paper focuses on solving least squares regression problem in the setting where some of the inputs are strategically corrupted. The authors show that certain special setting of the general problem, there exists an algorithm that achieves a provably optimal solution. Overall, the results are interesting and target an improved problems. The main concerns are regarding the significance of the results (as the setting targeted here seems to be fairly narrow) and providing a proper context wrt previous work (this hopefully can be remedied in the revision).